# Assessment of the Phytochemical Profile, Antioxidant Capacity, and Hepatoprotective Effect of *Andrographis paniculata* against CCl_4_-Induced Liver Dysfunction in Wistar Albino Rats

**DOI:** 10.3390/medicina59071260

**Published:** 2023-07-06

**Authors:** Syed Kashif Ali, Hafiz A. Makeen, Gulrana Khuwaja, Hassan A. Alhazmi, Mukul Sharma, Afraim Koty, Islam Mazahirul, Humaira Parveen, Asaduddin Mohammed, Sayeed Mukhtar, Mohammad Firoz Alam

**Affiliations:** 1Department of Chemistry, Faculty of Science, Jazan University, P.O. Box 114, Jazan 45142, Saudi Arabia; skali@jazanu.edu.sa; 2Department of Clinical Pharmacy, College of Pharmacy, Jazan University, P.O. Box 114, Jazan 45142, Saudi Arabia; hafiz@jazanu.edu.sa; 3Department of Pharmaceutical Chemistry and Pharmacognosy, College of Pharmacy, Jazan University, Jazan 45142, Saudi Arabia; gkhuwaja@jazanu.edu.sa (G.K.); haalhazmi@jazanu.edu.sa (H.A.A.); 4Center of Environmental Research and Studies, Jazan University, Jazan 42145, Saudi Arabia; mukulsharma@jazanu.edu.sa; 5Department of Chemistry, College of Science, Mahliya Jazan University, Jazan 45142, Saudi Arabia; akoty4@gmail.com; 6Department of Biology, Faculty of Science, Jazan University, Jazan 45142, Saudi Arabia; mohammed@jazanu.edu.sa; 7Department of Chemistry, Faculty of Science, University of Tabuk, Tabuk 71491, Saudi Arabia; h.nabi@ut.edu.sa (H.P.); snoor@ut.edu.sa (S.M.); 8Department of Pharmacology and Toxicology, College of Pharmacy, Jazan University, Jazan 45142, Saudi Arabia

**Keywords:** *Andrographis paniculata*, phytochemical profile, antioxidant activity, liver injury, hepatoprotective activity, CCl_4_

## Abstract

Recent studies have highlighted the necessity to thoroughly evaluate medicinal plants due to their therapeutic potential. The current study delves into the phytochemical profile, antioxidant capacity, and hepatoprotective effect of *Andrographis paniculata.* The investigation specifically targets its effectiveness in mitigating liver dysfunction induced by carbon tetrachloride (CCl_4_) in Wistar albino rats, aiming to uncover its promising role as a natural remedy for liver-related ailments. *A. paniculata* leaf extract was screened for phytoconstituents and antioxidant and hepatoprotective effects in Wistar albino rats against CCl_4_-induced liver dysfunction. Phytochemical analysis revealed the presence of flavonoids, alkaloids, and phenolic compounds in all extracts. The phenolic concentration ranged from 10.23 to 19.52 mg gallic acid per gram of the sample, while the highest flavonoid concentration was found in the ethanol fraction (8.27 mg rutin equivalents per gram). The antioxidant activity varied from 10.23 to 62.23. GC-MS analysis identified several phytochemicals including octadecanoic acid, stigmasterol, phenanthrenecarboxylic acid, and others. Effects of the ethanol extract of *A. paniculata* were evaluated in four groups of animals. Biochemical estimations of serum glutamine oxaloacetate transaminase, serum glutamine pyruvate transaminase, and serum bilirubin were significantly higher (*p* < 0.05) in the CCl_4_-treated group. Treatment with 300 mg/kg b.w. of the ethanol extract of *A. paniculata* significantly (*p* < 0.05) decreased these serum enzymes. Lipid peroxidation levels in carbon tetrachloride-treated animals showed a substantial (*p* < 0.05) rise when compared to untreated animals, while the lipid peroxidation levels were considerably (*p* < 0.05) reduced after treatment with ethanol extract at 300 mg/kg b.w. Liver biochemical catalase activities were significantly reduced in the carbon tetrachloride-treated animals. The results of this study conclusively demonstrate that *A. paniculata* extracts are a rich source of phytochemicals and possess significant antioxidant, free radical scavenging, and hepatoprotective properties.

## 1. Introduction

The WHO recognizes the importance of medicinal plants as a source of valuable therapeutic compounds for the treatment of various diseases and ailments [1,2]. These plants contain bioactive compounds that have therapeutic properties and are used in traditional medicine systems worldwide [3,4]. Scientific studies have confirmed the pharmacological activities of many medicinal plants (*Zingeber officinale*, *Nigella sativa, Capsicum annuum, Curcuma longa*) and their bioactive compounds. These compounds have been found to possess antioxidant, anti-inflammatory, antimicrobial, antiviral, antitumor, and other therapeutic effects [5].

The use of medicinal plants has gained renewed interest in recent years, and many researchers are focusing on developing new drugs and therapies based on these natural compounds. The study of medicinal plants involves the identification, extraction, and characterization of bioactive compounds and the evaluation of their pharmacological activities [6]. The use of medicinal plants and their bioactive compounds has several advantages over synthetic drugs, including their low toxicity, better tolerability, and fewer side effects [7]. Additionally, medicinal plants can be a source of affordable and accessible healthcare, particularly in low-income countries where modern medicines may not be readily available or affordable.

*A. paniculata* is an indigenous plant found in Taiwan, China, India, and Sri Lanka. It is prevalent in tropical and subtropical regions of Asia, including Southeast Asia, as well as in countries such as Cambodia, the Caribbean Islands, and Indonesia. Notably, it is used in traditional medicine systems such as Ayurveda, Siddha, Unani, and homeopathy to treat various ailments, including constipation, stomach gas, worms, and liver diseases The hepatoprotective properties of Kalmegh have been reported in studies conducted by Shankar et al. (2012) and Sanjhuta et al. (2008) [8,9].

The chemical composition of *A. paniculata* includes diterpenes, lactones, flavonoids, alkenes, ketones, aldehydes, and various active compounds. Andrographolide, also known as kalmegin, dioxyandrographolide, neoandrographolide, and dihydroandrographolide, is predominantly found in the leaves and contributes to their bitter taste. The leaves and stems of the plant contain flavonoids, gums, mucilages, and tannins [10]. The therapeutic value of these chemicals is determined by their presence and function in the human body. High-pressure chromatography can be employed to identify and standardize the main active ingredients of *A. paniculata* [11,12,13]. Andrographolide has been found to be present in the highest concentration (2.39% *w*/*w*) in the leaves, while the seeds contain the lowest concentration [14].

Phytosteroids identified in *A. paniculata* include β-sitosterol, stigmasterol, campesterol, and ergosterol. These constituents exhibit antioxidant properties, as demonstrated by studies investigating their effects on mitochondrial electron chain complexes and nitric oxide levels in the brain of rats [15]. Various bioactive compounds have been identified in *A. paniculata*, including andrographolide, neoandrographolide, andrographiside, andropanoside, and flavonoids, which have been reported to possess pharmacological activities [16]. The phytochemical composition of *A. paniculata* has been investigated through GC-MS analysis by Gupta and Kumar (2017) and Dulara et al. (2019) [17,18].

Among the bioactive compounds found in *A. paniculata*, andrographolide is one of the most extensively studied for its therapeutic properties. It exhibits anti-inflammatory, antioxidant, antimicrobial, antiviral, antitumor, hepatoprotective, cardioprotective, and neuroprotective activities [15]. Other bioactive compounds identified in Kalmegh include neoandrographolide, deoxyandrographolide, andrographiside, andropanoside, and andrograpanin. Additionally, flavonoids such as quercetin, kaempferol, and apigenin contribute to its therapeutic properties.

In recent years, *A. paniculata* has gained significant attention for its potent antioxidant activity, which has been extensively studied in both in vitro and in vivo models [19]. Numerous studies have reported the antioxidant activity of *A. paniculata* and its bioactive compounds. This antioxidant activity makes *A. paniculata* a valuable medicinal plant for the development of new drugs and therapies targeting health conditions associated with oxidative stress. The antioxidant effects of *A. paniculata* are believed to be mediated through various mechanisms, including modulation of the body’s antioxidant defense system, inhibition of inflammation, and modulation of the immune system. In vitro studies have demonstrated the plant’s ability to scavenge free radicals, inhibit lipid peroxidation, and enhance the activity of antioxidant enzymes such as superoxide dismutase, catalase, glutathione peroxidase, and glutathione [20].

*A. paniculata*, a widely recognized medicinal herb, exhibits significant potential as an herbal medication. However, to establish its efficacy, a comprehensive comparative study is essential to explore its phytochemical composition, bioactive constituents, antioxidant potential, and hepatoprotective properties. This research aims to assess extracts of *A. paniculata* to identify the presence of phytoconstituents and characterize the bioactive compounds within crude extracts using organic solvents. Additionally, gas chromatography–mass spectrometry (GC-MS) was employed for chemical profiling, enabling a comprehensive analysis of its phytochemical composition. The primary focus of this investigation was to evaluate the effectiveness of *A. paniculata* in ameliorating liver dysfunction induced by CCl_4_ in Wistar albino rats, shedding light on its potential as a natural remedy for liver-related disorders.

## 2. Methods and Material

### 2.1. Plant Material

Fresh, healthy leaves of *A. paniculata* were picked and recognized based on descriptions in the literature, herbarium analysis was used to confirm their authenticity, and the plant was verified. A voucher specimen (No. 158 AP 232/02) was deposited at the Herbarium of the Institute of Foreign Trade and Management (IFTM) University, Moradabad, India. The leaves were cleaned and washed with distilled water before being dried at room temperature in the shade. *A. paniculata* leaves were further dried in an oven (Thermo Fisher VT 6025; Thermo Electron Corporation, Asheville, NC, USA) at 50 °C, crushed by a grinder, and stored in airtight polythene bags. The dried powder was brought during summer vacation to the main laboratory of the College of Pharmacy, Jazan University for further analysis [21] 1981).

### 2.2. Animal Handling

Male Wistar albino rats reared in the animal house of the College of Pharmacy, Jazan University, Saudi Arabia were used for this study. The animals were housed in typical laboratory settings, including a constant room temperature (27 °C), a twelve-hour light cycle, and a 12 h dark cycle for the duration of the experiments. The rats were supplied with water ad libitum and fed on standard rodent pellets [22]. Ethical considerations and protocols for the care and use of experimental subjects were followed throughout the course of the investigation. The protocols utilized in the trials were evaluated and approved by the Institutional Animal Ethics Committee at the College of Pharmacy, Jazan University, Saudi Arabia with reference no COP/IAEC/2507/1438, dated: 2 December 2016).

### 2.3. Plant Extraction Method

Extraction of leaves and phytochemical analysis were carried out at the Department of Pharmaceutical Chemistry, College of Pharmacy, Jazan University, Saudi Arabia. Using a mortar and pestle, leaves were crushed into powder. The dried powder (450 g) was manually blended, dissolved (macerated) in 3.0 L of ethanol, and then placed in a shaker for three days. To remove contaminants, ethanol extracts were run down in silica gel column chromatography (mesh size 200–400) before being collected in clean flasks. A rotary evaporator already set at 50 °C was used to concentrate the various solvent fractions, i.e., ethanol, ethyl acetate, petroleum ether, hexane, dichloromethane (DCM). To create a stock solution for subsequent investigations, each fraction was reconstituted in a minimal amount of dimethyl sulfoxide. From the stock solution, five concentrations of each extract (100 µg mL^−1^, 200 µg mL^−1^, 300 µg mL^−1^, 400 µg mL^−1^, 500 µg mL^−1^) were prepared [23].

### 2.4. Phytochemical Analysis

#### 2.4.1. Qualitative Analysis of Phytoconstituents

The phytochemical screening method is commonly used to identify the presence of various classes of plant-derived chemicals or secondary metabolites in a given plant extract. In this particular case, the fraction of the dry extract of *A. paniculata* was subjected to phytochemical screening. During the screening, several classes of phytochemicals were tested including flavonoids, tannins, triterpenoids, steroids, alkaloids, and quinones. The screening process involved a series of chemical tests that are specific to each class of phytochemical.

#### 2.4.2. Estimation of Flavonoids

Flavonoids were detected as per a previously described method (Pandey et al., 2019) [24]. Briefly, in the plant extract, sulfuric acid was concentrated and applied to the extract. Anthocyanins are visible as a yellowish orange color, flavones are visible as yellow to orange hues, and flavonones are seen as orange to red.

#### 2.4.3. Estimation of Tannin

Tannin was detected as per the method previously described by Pandey et al. (2019) [24]. Briefly, the dried powdered sample weighed around 0.5 g, which was heated in 20 mL of water in a test tube before being filtered. A few drops of 0.1% ferric chloride were added, and the coloration was checked for brownish green or blue-blackness.

#### 2.4.4. Estimation of Terpenoids

Terpenoids were detected as per the method previously described by Pandey et al. (2019) [24]. Briefly, in 2 mL of chloroform, 5 g of each extract was mixed, and then a layer of concentrated H_2_SO_4_ (3 mL) was carefully added. To demonstrate that terpenoids were present, a reddish brown coloration of the interface was created.

#### 2.4.5. Estimation of Steroids

Steroids were detected as per a previously described method [25]. Briefly, one gram of extract was dissolved in a few drops of acetic acid and acetic aldehyde, warmed and cooled under running water, and a drop of sulfuric acid was applied along the test tube’s edges. Steroids are present when a reddish brown ring forms at the contact.

#### 2.4.6. Estimation of Alkaloids

To test for the presence of alkaloids in a plant extract, 0.5 g of the extract was mixed with 1 mL of 1% hydrochloric acid and heated. The mixture was then filtered, and 2 mL of the filtrate was taken and treated with Mayer’s reagent. The appearance of turbidity or precipitation, as well as a green color, indicated the presence of alkaloids in the extract [25].

#### 2.4.7. Estimation of Quinone

Quinones were detected as per the method previously described by Rajesh et al. (2014). Briefly, 1 mL of the leaf extract was mixed with 1 mL of concentrated sulfuric acid. If a red color is produced, it indicates the presence of quinones [25].

#### 2.4.8. Quantitative Estimation of Phenolic and Flavonoid Content

With a few minor modifications, the Folin–Ciocalteu reagent was employed to determine the TPC of leaf extracts. Gallic acid was used as standard. At 760 nm, absorbance was measured. Gallic acid equivalents were used to measure TPC in milligrams per gram of dry extract (mg GAE g^−1^). The total flavonoid concentration of various leaf extract solvent fractions was calculated. Using the colorimetric approach, TFC was estimated by obtaining the absorbance at 510 nm [19]. The standard reference was rutin. The amount of TFC was calculated as milligrams of rutin equivalents per gram of dry extract (mg RE g^−1^).

#### 2.4.9. Antioxidant Activity of Different Extracts

We measured the antioxidant activity of various solvent fractions and calculated the percentage inhibition of free radical scavenging activity. According to a previously reported procedure, the 2,2-diphenyl-1-picrylhydrazyl (DPPH) free radical scavenging assay was determined [19]. By measuring the reaction mixtures’ absorbance at 517 nm using a UV/visible spectrophotometer, the DPPH free radical was measured. Ascorbic acid served as the accepted standard for the study. Each test sample’s DPPH free radical scavenging % was estimated using the formula:$$\%\;\mathrm{Free}\;\mathrm{radical}\;\mathrm{scavenging}=\frac{\mathrm{Xc}-\mathrm{Xs}}{\mathrm{Xc}}\times 100$$
where:
Xc = Absorbance of control;Xs = Absorbance of the sample.

### 2.5. Hepatoprotective Role of Ethanol Extracts of Leaves

Male albino Wistar rats aged 2 to 3 months and 200 to 250 g in weight were used and maintained at room temperature. The oral toxicity test was conducted in accordance with OECD guideline and test no 423. All animals received the leaf extract orally, and at three-hour intervals their health and behavior were monitored [26]. After 48 h, the animals were monitored for any mortality. According to the published methodology, the acute oral toxicity of the extract in male albino Wistar rats was assessed. The carbon tetrachloride (CCl_4_) model was used to examine the plant extracts’ hepatoprotective potential. Animals were divided into four groups: Group I—vehicle control without CCl_4_ or ethanol extract, Group II—plant leaves’ ethanol extract only (300 mg/kg b.w), Group III—carbon tetrachloride (CCl_4_) at 0.1 mL/kg b.w., Group IV—treatment group with CCl_4_ (0.1 mL/kg body weight i.p) and extract at 300 mg/kg b.w. given by oral route for up to 18 days. The animals were dissected under ether anesthesia after 24 h after the last treatment. After an overnight fast, blood was drawn from the hearts of the rats in each group through cardiac puncture, which was then placed on a stand with previously labeled centrifuging tubes and left to clot for 30 min at room temperature. Centrifugation was used to separate the serum for 15 min at 3000 rpm. A fraction of the separated serum was utilized to estimate certain biochemical parameters, and 10% of the liver was homogenized and used to assess the biochemistry of the liver. Serum biochemicals such as serum glutamine pyruvate transaminase (SGPT or ALT), serum glutamine oxaloacetate transaminase (SGOT or AST), serum alkaline phosphatase (SALP), g-glutamate transpeptidase (GGTP), serum total bilirubin (TB), and total protein (TP) content were estimated using reported methods (Reitman and Frankel, 1957; Kind and King, 1954; Mallay and Evelyn, 1951; Szaszi, 1969; Lowery et al., 1951) [27,28,29,30,31]. Liver biochemical tests of lipid peroxidation (LPO) and catalase (CAT) were carried out as per methods described earlier (Devasagayam and Tarachand, 1987 and Kakkar et al., 1984) [32,33].

### 2.6. GC-MS Analysis

Ethanolic extract was subjected to GC-MS analysis to identify compounds responsible for antioxidant and hepatoprotective potential. The Shimadzu GC-17A gas chromatograph equipped with a flame ionization detector (FID) and an autosampler was used for the gas chromatography (GC) study. Fused silica capillary column OV-1, DB-1 (30 m × 0.53 mm, 0.5 m film thickness), programmed from 75–240 °C, was used. Initial temperature was 75 °C with 1 min hold time, ramping rate was 5 °C/min to 240 °C with 5 min holding time. Helium was used as the carrier gas, flowing at a rate of 1.0 mL min^−1^. The transmission line was 300 °C, the ion source for the electron impact (EI) was 220 °C, and the electron energy was 70 eV. The GC ran for 39 min in total. The sample preparation involved transferring 50 µL of the extract into a polypropylene tube and diluting it with 500 µL of ethyl acetate. After vortexing, the mixture was transferred to a GC vial, and 1 µL of the sample was injected into the GC-MS instrument. To identify the individual volatile constituents, their mass spectra were compared with those in the National Institute of Standards and Technology (NIST) library. The concentration of each volatile compound was expressed as a percentage relative to the total peak area obtained from the GC-MS analysis of the sample [34].

### 2.7. Statistical Analysis

The statistical analysis was carried out using Minitab 16 statistical software (Brandon Court Unit E1-E2, Progress Way, Coventry CV3 2TE, UK) by one-way ANOVA followed by Tukey’s post hoc test to ascertain statistical significances. The results were compared among each other and presented as ± standard error of the mean (SEM). The minimum criterion for the results to be statistically significant was set as a value of *p* < 0.05.

## 3. Results

### 3.1. Qualitative Assessment of Phytochemicals

The qualitative assessment of phytochemicals such as flavonoids, alkaloids and phenolic compounds found them to be present in all extracts tested in February 2022. Tannin was present in ethanolic extracts, however, it was absent in ethyl acetate, petroleum ether, hexane and DCM extracts. Triterpenoid was present in ethanol, ethyl acetate, and hexane extracts. Data on phytochemical constituents present in ethanolic extracts are summarized in Table 1.

### 3.2. Quantitative Assessment of Phenolic Content

Folin–Ciocalteu reagent was employed to determine the TPC of leaf extracts. All solvent fractions of the dried powder from extracts of fresh, healthy leaves of *A. paniculata* were calculated for phenolic content. The ethanol fraction had the highest level of phenolic content (mg GAE g^−1^) followed by ethyl acetate, hexane, petroleum ether, and DCM. The range of the phenolic level was 10.23 mg GAE g^−1^ to 19.52 mg GAE g^−1^ (Figure 1).

### 3.3. Quantitative Assessment of Flavonoid Content

The total flavonoid concentration of various leaf extract solvent fractions was calculated using the colorimetric approach and TFC was estimated by obtaining the absorbance at 415 nm. All solvent fractions of the dried powder from extracts of fresh, healthy leaves of *A. paniculata* were calculated for flavonoid content. The ethanol fraction had the highest level of flavonoid content (mg RE g^−1^) followed by DCM, hexane, ethyl acetate, and petroleum ether. The range of the flavonoid level was 3.34 mg RE g^−1^ to 8.27 mg RE g^−1^ (Figure 2).

### 3.4. Antioxidant Potential

All extracts were diluted to prepare a concentration from 500 µg mL^−1^ to 100 µg mL^−1^. In all solvent fractions, 2,2-diphenyl-1-picrylhydrazyl (DPPH) free radical scavenging activity was calculated. Inhibition percentages were measured in the range of 10.23 to 62.23. Table 2 provides a summary of the data for all solvent fractions.

### 3.5. Hepatoprotective Role of Ethanolic Extracts

The oral toxicity test was conducted in accordance with OECD guideline 423. Before conducting the main experiment, ethanolic extracts were checked for preliminary toxicity. The extracts were determined to be non-toxic up to a level of 300 mg/kg body weight. Since no mortality or major behavioral alterations were noticed at 300 mg/kg, this dose was considered for further experiments. Effects of ethanol extract of *A. paniculata* were checked in four groups of animals. Serum biochemical parameters such as SGPT (IU/L), SGOT (IU/L, ALP (IU/L), total bilirubin (mg/dL), total protein (mg/dL), and GGTP (µ/L) were checked in serum samples. Results of effects of ethanol extract of *A. paniculata* on biochemical parameters in rats with carbon tetrachloride-induced hepatotoxicity are summarized in Table 3. ALP, g-glutamate transpeptidase, SGOT, SGPT, and serum bilirubin were all significantly (*p* < 0.05) enhanced. Comparing the CCl_4_-treated group to the vehicle control group, the total protein level dropped, indicating liver injury. Treatment with 300 mg/kg b.w. of the ethanol extract of *A. paniculata* significantly (*p* < 0.05) decreased the SGOT, SGPT, SALP, and glutamate transpeptidase (GGTP) levels. The levels of SGOT, SGPT, SALP, g-glutamate transpeptidase, and bilirubin were significantly reduced by ethanolic extract. Liver biochemical parameters such as LPO (nM/mg protein) and catalase (U/mg protein) were also tested. LPO levels in CCl_4_-treated animals showed a substantial (*p* < 0.05) rise. When compared to CCl_4_-treated and untreated animals, the LPO levels were considerably (*p* < 0.05) reduced after treatment with ethanol extract at 300 mg/kg b.w. Liver biochemical catalase activities were significantly reduced in the CCl_4_-treated animal group and after treatment with an ethanol extract of 300 mg/kg b.w. Catalase activities were significantly (*p* < 0.05) elevated toward normal values as compared to the CCl_4_-treated animal group as well as the vehicle control group.

### 3.6. GC-MS Analysis of Ethanolic Extract

The results of GC-MS analysis of ethanolic extract are summarized in Table 4. The relative quantity of the chemical components present in the extract of *A. paniculata* was expressed as a percentage based on the peak area created in the chromatogram after about 50 µL of each sample was injected. By comparing the GC retention durations of *A. paniculata* with GC retention times, the components of *A. paniculata* were detected. The NIST library database was used to estimate the bioactive compounds in the ethanolic extracts. Data of compounds detected in GC-MS analysis are based on the number of peaks, retention time, and % area of the peak. Phytochemicals such as octadecanoic acid, stigmasterol, phenanthrenecarboxylic acid 7-etheny, l-9-octadecenoic acid, geranyl-α-terpinene, 13,15-octacosadiyne, methyl 8,10-octadecadiynoate, and Stigmast-5-en-3-ol, octadecanoate, (3beta) have been observed (Figure 3).

## 4. Discussion

Despite the widespread use of *A. paniculata* as a medicinal herb, there is a need for comprehensive and in-depth comparative research on its phytochemicals, bioactive components, antioxidant capacity, and hepatoprotective action to establish its efficacy as an effective herbal medicine. In light of this, the present study aimed to analyze extracts of *A. paniculata* to identify and characterize the bioactive chemicals present within the crude extracts, which were prepared using organic solvents and subjected to chemical profiling using gas chromatography–mass spectrometry. By focusing on the presence of phytoconstituents, this research endeavored to gain a better understanding of the medicinal properties of *A. paniculata* and its potential as a therapeutic agent. We reported the phytochemical constituents, antioxidant potential, and hepatoprotective role of extracts of *A. paniculata*. Phytochemical flavonoids, alkaloids, and phenolic compounds were found to be present in all extracts tested. Tannin was present in ethanolic extracts, however, it was absent in ethyl acetate, petroleum ether, hexane, and DCM extracts.

Flavonoids and phenolic compounds are a diverse group of secondary metabolites commonly found in plants. They are known for their antioxidant and medicinal properties. The solubility of these compounds depends on their chemical structure, specifically the presence of hydroxyl groups (-OH) and aromatic rings.

In polar solvents, such as ethanol and ethyl acetate, which have high dielectric constants, the hydroxyl groups and other polar functional groups of flavonoids and phenolic compounds readily interact with the solvent molecules through hydrogen bonding and dipole–dipole interactions. This allows for effective extraction of these compounds from the plant material. On the other hand, in non-polar solvents such as petroleum ether, hexane, and dichloromethane, the hydroxyl groups of flavonoids and phenolic compounds have limited solubility due to the non-polar nature of these solvents. However, the presence of aromatic rings in these compounds allows for some degree of solubility in non-polar solvents through non-polar interactions, such as van der Waals forces and hydrophobic interactions. Therefore, the presence of flavonoids and phenolic compounds in both polar and non-polar solvents can be attributed to the different types of interactions that these compounds can undergo based on the solvent’s polarity.

The GC-MS analysis of the ethanolic extract of *A. paniculata* revealed the presence of bioactive compounds such as octadecanoic acid, stigmasterol, phenanthrenecarboxylic acid, geranyl-α-terpinene, 13,15-octacosadiyne, G. methyl 8,10-octadecadiynoate, and stigmast-5-EN-3-OL (3 β). Octadecanoic acid, a saturated fatty acid, has been reported to possess antimicrobial, anti-inflammatory, and antioxidant properties [35]. Stigmasterol, a phytosterol, shows potential anti-inflammatory and anticancer effects [36]. Phenanthrenecarboxylic acid, a phenolic compound, exhibits antioxidant and antimicrobial activities [37]. Geranyl-α-terpinene, belonging to the class of terpenes, is known for its diverse biological activities, including antimicrobial, antioxidant, and anti-inflammatory properties [38]. 13,15-octacosadiyne, a long-chain alkyne, possesses anti-inflammatory and antimicrobial properties. G. methyl 8,10-octadecadiynoate, another alkyne compound, shows potential anticancer activity. Stigmast-5-EN-3-OL (3 β), a phytosterol, is recognized for its anti-inflammatory and antioxidant effects [39] (Arora et al., 2018).

Studies conducted in different parts of the world clearly support that bioactive substances extracted using organic solvents from *A. paniculata* leaves contain phytochemical components with potential antioxidant activity (Das and Srivastav, 2014; Tiwari, 2017) [40,41]. In a recent study, Hamid et al. (2023) evaluated various extracts of *A. paniculata* for antioxidant properties. According to these findings, it has been concluded that andrographolide and *A. paniculata* leaf extract are good sources of biochemical components that can be utilized to treat skin discoloration [15].

In our study, the effects of ethanol extract of *A. paniculata* were checked in four groups of animals. Table 3 provides the values of various biochemical parameters measured in the four different animal groups. The present finding showed that the values of liver function markers such as serum glutamic pyruvic transaminase (SGPT), serum glutamic-oxaloacetic transaminase (SGOT), alkaline phosphatase (ALP), total bilirubin, and gamma-glutamyl transpeptidase (GGTP) were significantly increased in CCl_4_-administered group as compared to the control group. These parameters are markers of liver damage, and the increase in their levels suggests that CCl_4_ administration caused liver injury in these animals. The increase in total bilirubin, which is a breakdown product of hemoglobin, also suggests that CCl_4_ caused damage to the liver’s ability to process this compound. In contrast, the animals treated with leaf ethanol extract had levels of these parameters that were similar to the control group. This suggests that the extract did not cause any significant damage to the liver. Interestingly, when the ethanolic extract was given in combination with CCl_4_, the values of these parameters were significantly lower than in CCl_4_-administered animals, indicating that the extract might have some protective effect against CCl_4_-induced liver damage. The total protein levels were significantly lower in CCl_4_-administered animals, indicating that liver damage caused a reduction in protein synthesis. However, in the animals treated with only ethanolic extract, total protein levels were similar to the control group, suggesting that the extract did not cause any significant changes in protein synthesis.

The values of lipid peroxidation (LPO), which is a marker of oxidative stress, were significantly higher in CCl_4_-adminstered group as compared to the control group. This suggests that CCl_4_ administration caused oxidative damage to the liver. However, when the ethanolic extract was given in combination with CCl_4_, the values of LPO were significantly lower than in CCl_4_-administered animals, indicating that the extract might have some antioxidant effect. Finally, the antioxidant effects were confirmed by the analysis of catalase and the results revealed that the values of catalase, an antioxidant enzyme, were significantly lower in CCl_4_-administered animals, suggesting that CCl_4_ administration caused a reduction in the liver’s antioxidant capacity. However, in the animals treated with the ethanolic extract, catalase levels were similar to the control group, indicating that the extract did not cause any significant changes in antioxidant capacity.

Based on the present findings, it has been concluded that ethanol extracts from *A. paniculata* have good antioxidant and hepatoprotective properties. However, further studies are needed to confirm these findings and understand the underlying mechanisms. A number of studies have demonstrated the hepatoprotective role of various extracts of *A. paniculata* by using different animal models to induced liver toxicity [42,43,44,45]. Rats with CCl_4_-induced liver impairment were used to study the impact of *A. paniculata* extract [26]. Physical and biological changes were used to gauge how well protected an area was. The physical and biochemical alterations in the liver caused by CCl_4_ were greatly reduced by pre-treating with extract. Its actions might help in reducing acute liver damage brought on by chemicals. It can be said that *A. paniculata* aqueous extract is almost significantly effective when used in traditional medicine [46].

All previous studies support our finding that ethanolic extract of *A. paniculata* leaves showed hepatoprotective effects against CCl_4_-induced liver toxicity. Previous studies also supported our findings that *A. paniculata* has a strong antioxidant properties. This study was unique because of the comprehensive study of *A. paniculata* active ingredients and its hepatoprotective effects due to antioxidant properties. This is a non-clinical study, hence it has the drawback of not being able to apply the findings directly to humans until extensive toxicology studies and clinical trials have been completed.

## 5. Conclusions

The present study focused on evaluating the phytochemical profile, antioxidant capacity, and hepatoprotective effect of *A. paniculata* extracts. The results revealed the presence of phytochemical constituents such as flavonoids, alkaloids, and phenolic compounds in leaf extracts. Based on these findings, it can be concluded that the ethanol extract of *A. paniculata* possesses hepatoprotective properties, prevents lipid peroxidation, and exhibits antioxidant effects. However, further studies are warranted to elucidate the underlying mechanisms such as inflammatory markers and histopathology and validate these findings.

## Figures and Tables

**Figure 1 medicina-59-01260-f001:**
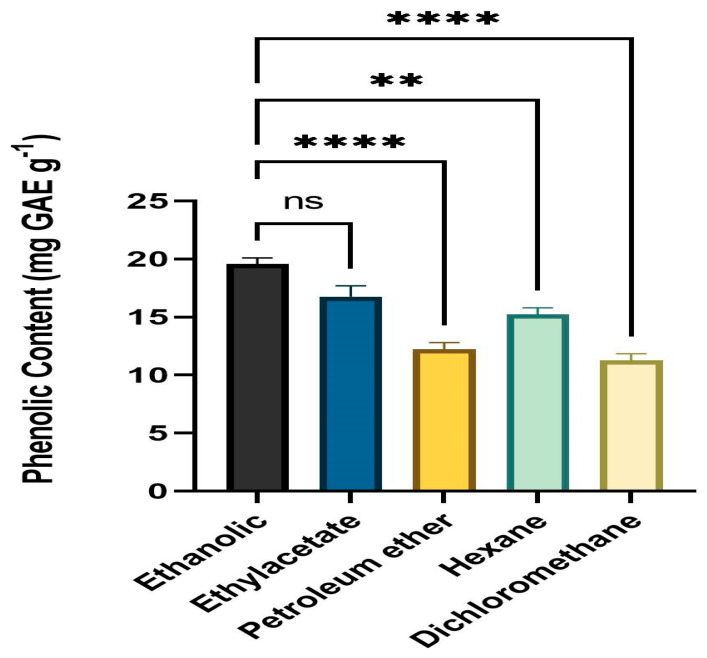
Phenolic content (mg GAE g^−1^) means with different star are significantly different at *p* < 0.05. ** *p* < 0.001 (Ethanol vs. Hexane), **** *p* < 0.0001 (Ethanol vs. Petroleum ether and Dichloromethane, ^ns^
*p* > 0.05 (Ethanol vs. Ethylacetate).

**Figure 2 medicina-59-01260-f002:**
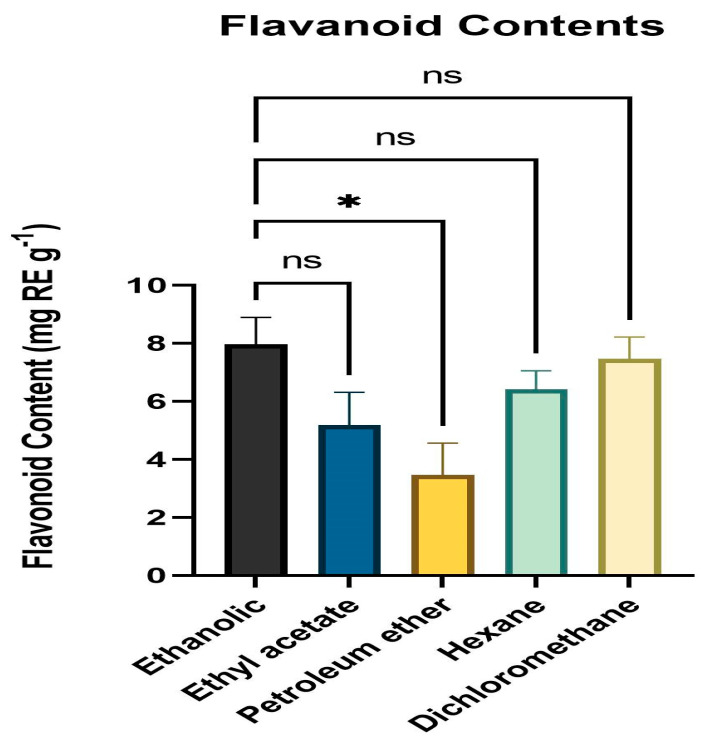
Flavonoid content (mg RE g^−1^) means with different letters are significantly different at *p* < 0.05. * *p* < 0.01 (Ethanol vs. Petroleum ether), ^ns^
*p* > 0.05 (Ethanol vs. Ethyl acetate, Hexane, Dichloromethane).

**Figure 3 medicina-59-01260-f003:**
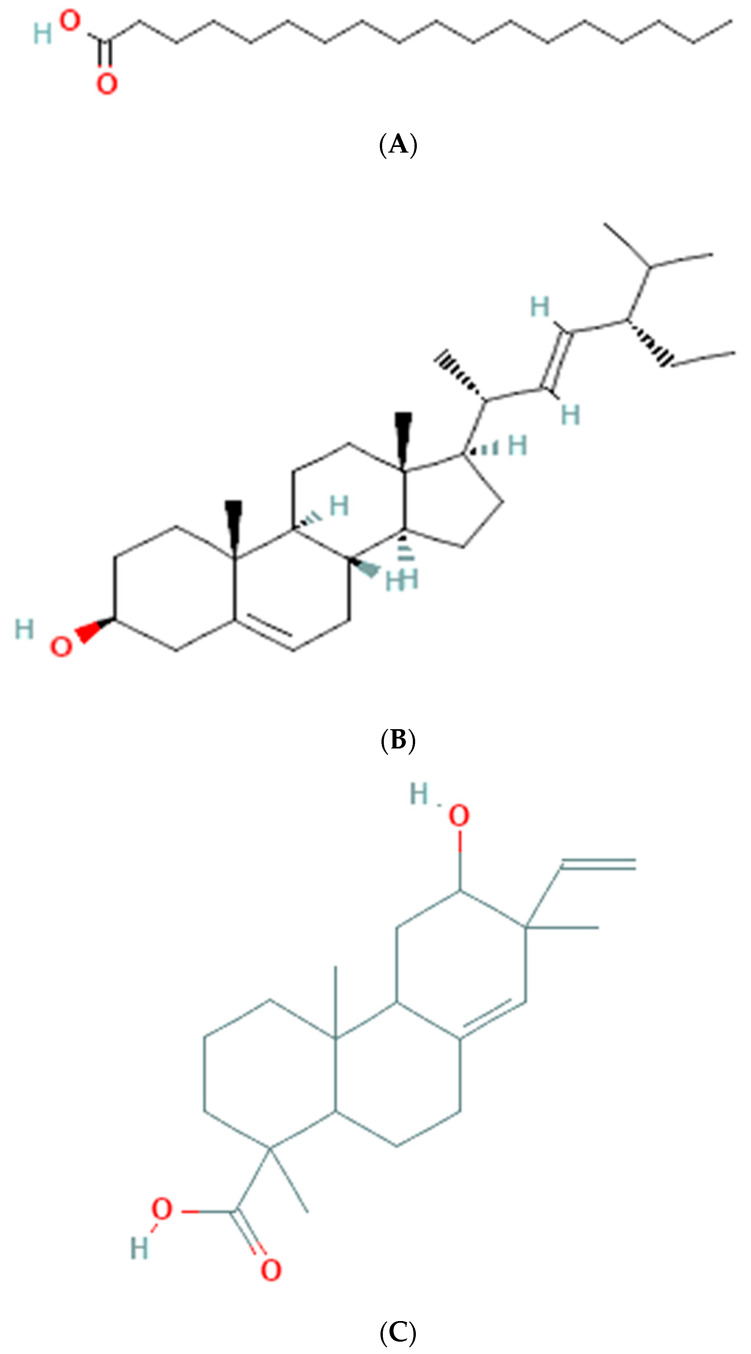
Molecular structure of phytoconstituents present in ethanolic leaf extract of Andrographis paniculate: (**A**) octadecanoic acid; (**B**) stigmasterol; (**C**) phenanthrene carboxylic acid; (**D**) 7-ethenyl 9-octadecenoic acid; (**E**) geranyl-α-terpinene; (**F**) 13,15-octacosadiyne; (**G**) methyl 8,10-octadecadiynoate; (**H**) stigmast-5-en-3-ol, octadecanoate, (3beta).

**Table 1 medicina-59-01260-t001:** Major phytochemicals of extracts of fresh, healthy leaves of *A. paniculata*.

Phytoconstituents	Organic Solvent Extracts				
	Ethanol	Ethyl Acetate	Petroleum Ether	Hexane	DCM
Flavonoids	+	+	+	+	+
Tannin	+	−	−	−	+
Terpenoids	+	+	−	+	+
Steroids	+	−	−	−	−
Alkaloids	+	+	+	+	+
Quinone	+	−	−	−	−
Phenolic compounds	+	+	+	+	+

+ (Present), − (Absent).

**Table 2 medicina-59-01260-t002:** Antioxidant potential of fresh, healthy leaves extract of *A. paniculata*.

Solvent Fractions	Concentrations of Extracts (µg mL^−1^)				
	100	200	300	400	500
Ethanolic	23.2 ± 3.2	43.1 ± 3.2	50.2 ± 1.3	55.5 ± 4.2	62.3 ± 2.3
Ethyl acetate	12.3 ± 2.1	23.3 ± 3.2	33.34 ± 2.4	42.2 ± 3.2	51.2 ± 4.2
Petroleum ether	9.2 ± 3.1	16.23 ± 4.2	22.34 ± 4.2	32.3 ± 4.1	45.32 ± 2.3
Hexane	16.3 ± 2.3	33.23 ± 1.5	42.12 ± 3.2	52.12 ± 4.1	61.4 ± 1.5
DCM	10.23 ± 2.1	18.34 ± 3.9	28.12 ± 3.1	39.2 ± 3.2	51.23 ± 1.3

Data represent the mean ± standard deviation with three replicates (*n* = 3).

**Table 3 medicina-59-01260-t003:** Effects of ethanol extract on biochemical parameters in rats against carbon tetrachloride-induced hepatotoxicity.

Biochemical Parameters	Animal Groups			
	Group I	Group II	Group III	Group IV
	Vehicle Control (0.0 mg/kg b.w)	Leaf Ethanol Extract (300 mg/kg b.w)	CCl_4_ (0.1 mL/kg b.w)	Leaf Extract + CCl_4_
SGPT (IU/L)	30.12 ± 3.1	35.23 ± 4.2 ^ns^	121.3 ± 4.5 ^a^	55.23 ± 1.5 ^b^
SGOT (IU/L)	41.2 ± 6.2	48.23 ± 4.9 ^ns^	119.12 ± 5.8 ^a^	52.23 ± 7.2 ^b^
ALP (IU/L)	9.8 ± 2.3	12.22 ± 4.3 ^ns^	38.23 ± 5.1 ^a^	18.23 ± 5.7 ^b^
Total bilirubin (mg/dL)	0.32 ± 0.02	0.41 ± 0.02 ^ns^	6.23 ± 0.02 ^a^	0.67 ± 0.02 ^b^
Total protein (mg/dL)	6.23 ± 1.2	6.78 ± 2.2 ^ns^	4.23 ± 1.4 ^a^	7.12 ± 2.1 ^b^
GGTP (µ/L)	71.23 ± 4.6	81.23 ± 5.3 ^ns^	160.34 ± 7.2 ^a^	90.23 ± 1.3 ^b^
LPO (nM/mg protein)	2.33 ± 0.03	2.44 ± 0.02 ^ns^	6.12 ± 0.05 ^a^	2.98 ± 0.03 ^b^
Catalase (U/mg protein)	32.12 ± 2.1	30.32 ± 1.9 ^ns^	14.23 ± 1.4 ^a^	27.23 ± 1.7 ^b^

These values represents the mean ± standard deviation; ^a^
*p* < 0.0001 represents the significant value between Group III vs. Group I, ^b^
*p* < 0.001 represents comparisons of the treatments of ethanolic extract such as Group IV vs. Group III, while ^ns^
*p* > 0.05 indicates no significant difference between Group II vs. Group I.

**Table 4 medicina-59-01260-t004:** GC-MS analysis of phytoconstituents present in ethanolic leaf extract of *A. paniculata*.

Peak No	R. Time	Area %	Molecular Weight	Molecular Formula	Name of the Compounds
1	18.122	1.21	284	C_18_H_36_O_2_	octadecanoic acid
2	32.211	3.23	412	C_29_H_48_O	stigmasterol
3	29.162	19.12	318	C_20_H_30_O_3_	phenanthrenecarboxylic acid, 7-ethenyl
4	16.212	6.32	282	C_18_H_34_O_2_	1-9-octadecenoic acid
5	27.312	2.12	272	C_20_H_32_	geranyl-α-terpinene
6	24.132	5.23	386	C_28_H_50_	13,15-octacosadiyne
7	27.132	1.23	290	C_19_H_30_O_2_	methyl 8,10-octadecadiynoate
8	32.253	7.54	414	C_29_H_50_O	stigmast-5-en-3-ol (3β)

## Data Availability

Suggested Data Availability Statements are available in section.

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
