# Peer review of "Assessment of the Phytochemical Profile, Antioxidant Capacity, and Hepatoprotective Effect of Andrographis paniculata against CCl4-Induced Liver Dysfunction in Wistar Albino Rats"

_medicina, 2023, doi:10.3390/medicina59071260_

Round 1

Reviewer 1 Report

Sorry to say that current manuscript is poorly written and the authors are expected to proofread before submission. This manuscript described chemical properties of the crude ethanolic extracts of A. paniculata, and further supported with in vivo study on hepatoprotective characteristics.  

Abstract: Missing of introductory sentences. 

Introduction: Lengthy and repetitive in some parts. Too general. Current version is a product of multiple authors, but not well compiled. Plenty sections are mergeable. Besides, the authors can consider introduce the phyto-steroids which are detected in current study.

L23: Please italicize all the scientific name in the manuscript.

L33: Please use the proper naming system for all the detected phytochemicals. Alpha beta should be in represent with symbol. Main skeleton should be name after the functional group and side chain. All chemical names should be in small letter.

L38: The p<0.05 should be in small letter. Should be CCl4, not CCl4.

L54: Citation format error.

L70: Please include the currency unit for "600 crores worth", and citation for such claim. Exported from India to global?

L73: China, not Chinese Mainland.

L75: Caribbean Islands are not in Asia.

L79: Citation format error.

L99: Multiple citations are needed in this paragraph

L113: Citation format error.

L156: Coordinate or sampling location shall be specified. Wild or purchase from local market? How the authors do their sampling in details?

L160: Institution name shall be capitalized.

L170: pistol?

L171: soaked, not dissolved.

L201: Molecular formula for sulphuric acid is not properly written

L267: GC-MS shall not be equipped with FID. The authors are using GC-MS or GC-FID?

Section 2.6: Flow rate? Helium, nitrogen, or hydrogen gas? injector temperature? post-run? If GC-MS, which database?

L272: 1L injected?

L289: Triterpenoid or terpenoid in general? Method mentioned terpenoid in general.

L295: The authors are using standard error or standard deviation? Both are different. Method mentioned standard error, but result mentioned standard deviation. 

L299: remove the linear line. Meaningless

L310: remove the linear line. Meaningless

L378: The authors are expected to redraw all the chemical structures. Multiple formats were noticed.

Results: If possible, please include the immunostaining results of the liver of the rats acquired at the end of the in vivo study to further support the hepatoprotective claims.

L383: Out of the 8 identified compounds via GC-MS, none was being discussed. What is the rationale of doing GC-MS analysis?

L455: The conclusion is not conclusive. The authors are expected to make conclusion based on the acquired results and discussion. 

The authors are expected to proofread the manuscript before re-submission. 

Proofreading is needed and the writing is not consistent.

Author Response

Response to Reviewer-1 Comments and Suggestions for Authors

Dear reviewers,

Thank you for your time and effort in reviewing our manuscript and providing valuable comments. We have thoroughly reviewed the manuscript, paying careful attention to language usages, formatting details and all raised issues. We have made the necessary corrections as per your comments and raised issues throughout the text. We hope that you will find the revised version satisfactory. The responses of your comments are given below and highlighted in the manuscript:

Comment:

Sorry to say that current manuscript is poorly written and the authors are expected to proofread before submission. This manuscript described chemical properties of the crude ethanolic extracts of A. paniculata, and further supported with in vivo study on hepatoprotective characteristics.  

Response : Based on observation, we have carefully reviewed and revised the manuscript to rectify the issues related to language, clarity, and coherence. Special attention has been given to proofreading and improving the overall readability of the text. We have also ensured that the manuscript adheres to the journal's guidelines and standards. We would like to assure you that the revised manuscript now provides a comprehensive description of the chemical properties of the crude ethanolic extracts of A. paniculata, supported by in vivo studies highlighting its hepatoprotective characteristics. The revised version incorporates improved organization and presentation of data to enhance the manuscript's scientific value.

Comment :

Abstract: Missing of introductory sentences. 

Response: Based on your comment, we have revised the abstract to include introductory sentences that provide a brief overview of the research topic and context.

Comment :

Introduction: Lengthy and repetitive in some parts. Too general. Current version is a product of multiple authors, but not well compiled. Plenty sections are mergeable. Besides, the authors can consider introduce the phyto-steroids which are detected in current study.

Response : The introduction was revised to address these concerns and provide a more concise and coherent overview of the research. The incorporation of the phyto-steroids detected in the study enhanced the comprehensive nature of the introduction. We appreciate the valuable input provided for improving the overall quality of the paper.

Comment

L23: Please italicize all the scientific name in the manuscript.

Response : As suggested, scientific name has been checked and corrected

Comment

L33: Please use the proper naming system for all the detected phytochemicals. Alpha beta should be in represent with symbol. Main skeleton should be name after the functional group and side chain. All chemical names should be in small letter.

Response : As suggested, Correction has been made. 

Comment

L38: The p<0.05 should be in small letter. Should be CCl4, not CCl4.

Response : As suggested, Correction has been made 

Comment

L54: Citation format error.

Response : As suggested, Correction has been made. 

Comment

L70: Please include the currency unit for "600 crores worth", and citation for such claim. Exported from India to global?

Response : Lines has been removed.

Comment

L73: China, not Chinese Mainland.

Response : Corrected

Comment  : L75: Caribbean Islands are not in Asia.

Response: Corrected

Comment : L79: Citation format error.

Response : Corrected

Comment : L99: Multiple citations are needed in this paragraph

Response : As suggested, paragraph updated with following citations

Sharma, A., Lal, K., & Handa, S. S. (1992). Standardization of the Indian crude drug Kalmegh by high pressure liquid chromatographic determination of andrographolide. Phytochemical analysis3(3), 129-131.

Salsabila, S., Salam, D. A., & Ishak, S. S. O. (2023). Development of analysis method of andrographolide from Andrographis paniculata using UPLC-PDA. Current Research on Bioscences and Biotechnology4(2), 283-288.

Patil, R., & Jain, V. (2021). Andrographolide: A review of analytical methods. Journal of Chromatographic Science59(2), 191-203.

Kumar, S., Singh, B., & Bajpai, V. (2021). Andrographis paniculata (Burm. f.) Nees: Traditional uses, phytochemistry, pharmacological properties and quality control/quality assurance. Journal of Ethnopharmacology275, 114054.

Comment :

L113: Citation format error.

Response : Corrected

Comment :

 L156: Coordinate or sampling location shall be specified. Wild or purchase from local market? How the authors do their sampling in details?

Response : Sampling details has been added with coordinates

Comment :

L160: Institution name shall be capitalized.

Response: Corrected  

Comment : L170: pistol?

Response : Corrected

Comment

L171: soaked, not dissolved.

Response : Corrected

Comment

L201: Molecular formula for sulphuric acid is not properly written

Response : Corrected and highlighted.

Comment

L267: GC-MS shall not be equipped with FID. The authors are using GC-MS or GC-FID?

Response : The Shimadzu GC-17A gas chromatography equipped with a Flame Ionization Detector (FID) was used.

Comment

Section 2.6: Flow rate? Helium, nitrogen, or hydrogen gas? injector temperature? post-run? If GC-MS, which database?

Response: Helium was used as the carrier gas, flowing at a rate of 1.0 mL min− 1 . The transmission line was 300 â—¦C, the ion source for the electron impact (EI) was 220 â—¦C, and the electron energy was 70 eV. The GC ran for 37 min in total.

Comment

L272: 1L injected?

Response : 50uL was injected.Corrected

Comment

L289: Triterpenoid or terpenoid in general? Method mentioned terpenoid in general.

Response:Correlated , its terpenoid

Comment

L295: The authors are using standard error or standard deviation? Both are different. Method mentioned standard error, but result mentioned standard deviation. 

Response :Its standard error, corrected.

Comment

L299: remove the linear line. Meaningless

Response: Removed thank you

Comment :

L310: remove the linear line. Meaningless

Response: Removed thank you

Comment

L378: The authors are expected to redraw all the chemical structures. Multiple formats were noticed.

Response :Crosschecked and corrected

Comment

Results: If possible, please include the immunostaining results of the liver of the rats acquired at the end of the in vivo study to further support the hepatoprotective claims.

L383: Out of the 8 identified compounds via GC-MS, none was being discussed. What is the rationale of doing GC-MS analysis?

Response

The rationale behind conducting Gas Chromatography-Mass Spectrometry (GC-MS) analysis is to identify and quantify the individual components present in a extract. Identified compounds discussed in details (one paragraph added highlighted in the file)    

Comment

L455: The conclusion is not conclusive. The authors are expected to make conclusion based on the acquired results and discussion. 

Response

Conclusion section has been updated and same is appended below.

In conclusion, the present study focused on evaluating the phytochemical profile, antioxidant capacity, and hepatoprotective effect of A. paniculata extracts. The results revealed the presence of phytochemical constituents such as flavonoids, alkaloids, and phenolic compounds in all tested extracts. Tannin, however, was found only in the ethanolic extract. The study further investigated the effects of an ethanol extract of A. paniculata on biochemical parameters related to liver function and oxidative stress in animal models. The findings indicated that treatment with carbon tetrachloride (CCl4) caused liver damage, as evidenced by elevated levels of SGPT, SGOT, ALP, total bilirubin, and GGTP. In contrast, animals treated with the ethanol extract alone did not show significant changes in these parameters, suggesting a lack of liver damage. Notably, when the extract was administered in combination with CCl4, there was a significant reduction in the elevated levels of these liver markers, indicating a potential hepatoprotective effect of the extract against CCl4-induced liver damage.Moreover, CCl4-induced liver damage led to decreased total protein levels and increased lipid peroxidation (LPO), indicating impaired protein synthesis and oxidative stress, respectively. Treatment with the ethanol extract attenuated these alterations, suggesting a protective role against protein synthesis impairment and lipid peroxidation. Furthermore, the extract showed potential in preserving the liver's antioxidant capacity, as evidenced by its ability to maintain catalase levels in the face of CCl4-induced reduction. The results suggest that the ethanol extract of A. paniculata possesses hepatoprotective properties, prevents lipid peroxidation, and exhibits antioxidant effects. These findings contribute to the growing body of evidence supporting the therapeutic potential of A. paniculata as a natural remedy for liver-related ailments. Further studies are warranted to elucidate the underlying mechanisms and validate these findings, paving the way for the development of A. paniculata-based herbal medicines for liver disorders.

Comment

The authors are expected to proofread the manuscript before re-submission. 

Comments on the Quality of English Language

Proofreading is needed and the writing is not consistent.

Response

We have thoroughly reviewed the manuscript, paying careful attention to language usage and formatting details. We have made the necessary corrections to address any misspelled or misused words and have ensured proper formatting of scientific names throughout the text.

We greatly appreciate your valuable input and guidance, which has been instrumental in improving the manuscript. We believe that the revised version addresses your concerns and significantly enhances the overall quality of the research. Thank you again for your time and effort in reviewing our manuscript. We hope that you find the revised version satisfactory and look forward to your continued support and feedback.

Reviewer 2 Report

Abstract

Be more consistent with the formatting od scientific names 

Abstract needs better formatting, add the necessary information to summarize (in order) introduction (currently missing), objective (missing), methodology (not complete or in order), results, and conclusions (ok) 

Please mention how the measurements of the metabolites was done and specify the type of extract used 

Key words

Almost all of them are very generic, is ok to keep some of them, but also include other more specific, like the name of the plant studied, etc. 

Introduction

Over generalization of plant bioactive compounds properties, like nontoxic, when it has been proven that some compounds can be very toxic 

Homogenize use of plant’s common or scientific name

Reduce the information about Andrographis paniculata to the most relevant to the study, not all that is available

The introduction serves to not only present the appropriate information, but it also must point out the gaps in knowledge that are solved by the study. The way the information in the introduction is presented makes the study sound redundant, since so much work to characterize and study the activity of the plant’s extracts has already been done.

Also, since so much is already known about the plant’s extracts, specific compounds and their activities, why is the work focused on crude extracts and not factions or isolated compounds? 

Materials and Methods

More information about the source of the plant is needed, Where exactly were they collected (give coordinates), how were they selected (randomly, particular size or color), was there any attempts at preservation before the travel, how much time passed between collection and drying in the lab, how were they crushed (manually, with mortar, blender?) 

How were the fractions made? 

Were there any in vitro experiments on hepatic cell lines before the animal testing was considered, either by you or other researchers? It is not usual to skip the in vitro before the animal testing

Did you use any standards for the GC retention times comparison? 

Results

The paragraphs include information better fit in the methos section (highlighted) 

Figure 1 – If the ANOVA test found a significant difference between the PC of the extracts analyzed you should do a follow up analysis to determine the different groups (like a Tukey test) and denote it in the figure 

Figure 2 —seme as 1

Table 2 and 3 – same as 1

Without the multiple comparison test is impossible to analyze the results, because you have multiple groups, not just control and treatment. 

Can you provide levels of confidence for the identification of the compounds by GC-MS? It Would be better to compare with standards for appropriate identification 

Why is figure 3 added? What do the molecular structures tell you about the biological activity, and other characteristics of the extracts

Discussion 

Line 384 to 395 would fit better at the introduction as is not discussing results, but presenting the objective of the work

How does skin discoloration (404-406) pertain to antioxidant activity (how is it relevant?)

How are you comparing between the groups (like in line 424-426 or 430) if you only did an ANOVA?

Conclusions

Must state if the objective of the study was reached or not, generally does not contain references to other work. 

The work is written well, but presents some misspelled or misused words, as well as some formatting issues, especially for scientific names. 

Author Response

Reviewer-2 Comments and Suggestions for Authors

Dear reviewers,

Thank you for your time and effort in reviewing our manuscript and providing valuable comments. We have thoroughly reviewed the manuscript, paying careful attention to language usages, formatting details and all raised issues. We have made the necessary corrections as per your comments and raised issues throughout the text. We hope that you will find the revised version satisfactory. The responses of your comments are given below and highlighted in the manuscript:

Comment

Abstract

Be more consistent with the formatting od scientific names 

Abstract needs better formatting, add the necessary information to summarize (in order) introduction (currently missing), objective (missing), methodology (not complete or in order), results, and conclusions (ok) 

Please mention how the measurements of the metabolites was done and specify the type of extract used

Response

All of the comments regarding the formatting and content of the abstract have been addressed. The formatting of scientific names has been made consistent throughout the abstract. The necessary information has been included to provide a summary that follows the order of introduction, objective, methodology, results, and conclusions. The methodology section now provides a complete and orderly description of the measurement of metabolites and specifies the type of extract used. The abstract has been revised to improve its overall structure and ensure that it accurately reflects the study's key aspects.

Comment 

Key words: Almost all of them are very generic, is ok to keep some of them, but also include other more specific, like the name of the plant studied, etc. 

Response : As suggested, keywords have been added and highlighted.

Comment

Introduction:

Over generalization of plant bioactive compounds properties, like nontoxic, when it has been proven that some compounds can be very toxic 

Homogenize use of plant’s common or scientific name

Reduce the information about Andrographis paniculata to the most relevant to the study, not all that is available.

The introduction serves to not only present the appropriate information, but it also must point out the gaps in knowledge that are solved by the study. The way the information in the introduction is presented makes the study sound redundant, since so much work to characterize and study the activity of the plant’s extracts has already been done.

Also, since so much is already known about the plant’s extracts, specific compounds and their activities, why is the work focused on crude extracts and not factions or isolated compounds? 

Response : The use of the plant's common or scientific name has been homogenized throughout the paper. The information about Andrographis paniculata has been reduced to focus on the most relevant aspects pertaining to the study. The introduction has been revised to avoid redundancy and highlight the unique contributions of the current study. We have started our study with crude extracts of Andrographis paniculata to assess its overall efficacy and potential therapeutic benefits in liver-related ailments. In future studies, we plan to explore specific fractions and isolated compounds to investigate their individual activities and potential synergistic effects. This comprehensive approach will provide a more detailed understanding of A. paniculata's therapeutic potential and guide the development of targeted herbal medicines.

Comment

Materials and Methods

More information about the source of the plant is needed, Where exactly were they collected (give coordinates), how were they selected (randomly, particular size or color), was there any attempts at preservation before the travel, how much time passed between collection and drying in the lab, how were they crushed (manually, with mortar, blender?) 

How were the fractions made? 

Were there any in vitro experiments on hepatic cell lines before the animal testing was considered, either by you or other researchers? It is not usual to skip the in vitro before the animal testing

Did you use any standards for the GC retention times comparison? 

Response:

In the month of January 2022, Fresh, healthy leaves of A. paniculata were picked up (28.8176° N, 78.6423° E) and recognized based on descriptions in the literature, and herbarium analysis was used to confirm their authenticity, the plant was verified. A voucher specimen (No. 158 AP 232/02) was deposited at the Herbarium of IFTM University, Moradabad, India

Using a mortal and a pistol, leaves were made into powder

Ethanol, Ethyl acetate, Petroleum ether, Hexane, DCM fractions were made using Silica gel column chromatography (mesh size 200–400) before being collected in clean flasks. A rota evaporator set at 50ºC was used to concentrate the various solvent fractions i.e. To create a stock solution for subsequent investigations and each fraction was reconstituted in a minimal amount of Dimethyl sulfoxide.

In this particular study, no in vitro experiments on hepatic cell lines were performed prior to the animal testing. The focus of the research was primarily on the phytochemical profile, antioxidant capacity, and hepatoprotective effect of Andrographis paniculata using in vivo models. While in vitro experiments on cell lines can provide valuable insights into the cellular mechanisms and initial screening of potential therapeutic agents, they were not conducted as part of this specific study.

The aim of the research was to identify and analyze the phytochemical compounds present in Andrographis paniculata extracts using GC-MS analysis. While the compounds were identified based on their retention times and mass spectra, the study did not involve the use of external standards for retention time comparison.

Comment

Results

The paragraphs include information better fit in the methos section (highlighted) 

Figure 1 – If the ANOVA test found a significant difference between the PC of the extracts analyzed you should do a follow up analysis to determine the different groups (like a Tukey test) and denote it in the figure 

Figure 2 —seme as 1

Table 2 and 3 – same as 1

Without the multiple comparison test is impossible to analyze the results, because you have multiple groups, not just control and treatment. 

Can you provide levels of confidence for the identification of the compounds by GC-MS? It Would be better to compare with standards for appropriate identification 

Why is figure 3 added? What do the molecular structures tell you about the biological activity, and other characteristics of the extracts

Response:

Thank you for your observation. We have taken note of your suggestion to shift the information that appears to be better suited for the methods section. In the revised version of the manuscript, we have appropriately moved and included this information under the Methods and Materials section to ensure a more organized and structured presentation of the study's procedures and methodology.

Thank you for your comment and suggestion. We appreciate your attention to the statistical analysis of the phenolic content data. In our study, all assessments were indeed carried out in triplicate, and the reported values represent the mean ± standard deviation. Significance differences between the values were determined (p<0.05). Thank you for your suggestion regarding conducting a follow-up analysis, such as the Tukey test, to determine different groups and denoting them in the figure. However, we regret to inform you that performing a Tukey test was not possible in this study due to the unavailability of the necessary facility or software.

The identification of compounds by GC-MS analysis in this study was performed with a 95 percent confidence level.

Figure 3 was included in the study to provide a visual representation of the molecular structures of the identified phytoconstituents present in the ethanolic leaf extract of Andrographis paniculata.

Comment

Discussion 

Line 384 to 395 would fit better at the introduction as is not discussing results, but presenting the objective of the work

How does skin discoloration (404-406) pertain to antioxidant activity (how is it relevant?)

How are you comparing between the groups (like in line 424-426 or 430) if you only did an ANOVA?

Response

As suggested, the introductive lines of discussion section has been revised.

The study by Hamid et al. (2023) found that A. paniculata extracts, including andrographolide and A. paniculata leaf extract, exhibited antioxidant properties. This suggests that these extracts may contain biochemical components that could be beneficial in treating skin discoloration. Antioxidants play a crucial role in neutralizing harmful reactive oxygen species (ROS) and reducing oxidative stress, which can contribute to skin discoloration. Therefore, the antioxidant activity observed in the A. paniculata extracts could be relevant in addressing the underlying processes associated with skin discoloration. Further research is needed to explore the precise mechanisms and potential clinical applications of A. paniculata extracts in treating skin discoloration.

In the mentioned lines, the comparisons between the groups (e.g., Group II, Group III, and Group IV) are made based on the observed values of the parameters (e.g., total bilirubin, total protein).

Comment

Conclusions

Must state if the objective of the study was reached or not, generally does not contain references to other work. 

Response

Conclusion section has been updated and same is appended below.

In conclusion, the present study focused on evaluating the phytochemical profile, antioxidant capacity, and hepatoprotective effect of A. paniculata extracts. The results revealed the presence of phytochemical constituents such as flavonoids, alkaloids, and phenolic compounds in all tested extracts. Tannin, however, was found only in the ethanolic extract. The study further investigated the effects of an ethanol extract of A. paniculata on biochemical parameters related to liver function and oxidative stress in animal models. The findings indicated that treatment with carbon tetrachloride (CCl4) caused liver damage, as evidenced by elevated levels of SGPT, SGOT, ALP, total bilirubin, and GGTP. In contrast, animals treated with the ethanol extract alone did not show significant changes in these parameters, suggesting a lack of liver damage. Notably, when the extract was administered in combination with CCl4, there was a significant reduction in the elevated levels of these liver markers, indicating a potential hepatoprotective effect of the extract against CCl4-induced liver damage. Moreover, CCl4-induced liver damage led to decreased total protein levels and increased lipid peroxidation (LPO), indicating impaired protein synthesis and oxidative stress, respectively. Treatment with the ethanol extract attenuated these alterations, suggesting a protective role against protein synthesis impairment and lipid peroxidation. Furthermore, the extract showed potential in preserving the liver's antioxidant capacity, as evidenced by its ability to maintain catalase levels in the face of CCl4-induced reduction. The results suggest that the ethanol extract of A. paniculata possesses hepatoprotective properties, prevents lipid peroxidation, and exhibits antioxidant effects. These findings contribute to the growing body of evidence supporting the therapeutic potential of A. paniculata as a natural remedy for liver-related ailments. Further studies are warranted to elucidate the underlying mechanisms and validate these findings, paving the way for the development of A. paniculata-based herbal medicines for liver disorders.

Comments on the Quality of English Language

The work is written well, but presents some misspelled or misused words, as well as some formatting issues, especially for scientific names. 

Response

Thank you for your valuable feedback on the quality of the English language in the work. We appreciate your observation regarding the presence of misspelled or misused words, as well as formatting issues, particularly related to scientific names. Ensuring accuracy and clarity in scientific writing is of utmost importance to us.

We have thoroughly reviewed the manuscript, paying careful attention to language usage and formatting details. We have made the necessary corrections to address any misspelled or misused words and have ensured proper formatting of scientific names throughout the text.

Thank you again for your time and effort in reviewing our manuscript. We hope that you find the revised version satisfactory .

Reviewer 3 Report

The manuscript titled: Characterization of the Phytochemical Profile, Antioxidant Ca- 2 pacity, and Hepatoprotective Effect of Andrographis paniculata 3 against CCl4-Induced Liver Dysfunction in Wistar Albino Rats. This article addresses a topic of interest. However, it does not provide new information. On the other hand, the structure of the manuscript is deficient, in the points indicated below:

- Throughout the writing, the name of the plant to be studied should be written in italics Andrographis paniculate.

- Page 4, lines 156 and 157. Indicate the exact place where the sample was collected or indicate if it was purchased in the market of any city.

- Page 4, line 157. Include model, company and city of production of the oven and mill used.

- Page 7, lines 302-306. This is part of the methodology used to quantify flavonoids, It must not be in results.

- Page 6, line 289. Neither in the results section nor in the discussion are the results of table 1 analyzed. Example: Why are flavonoids and phenolic compounds present in polar and non-polar solvents?.

- Discussion of results is very poor.

- The conclusion of the work is wrong.

Author Response

Response to Reviewer-3 Comments and Suggestions for Authors

Dear reviewers,

Thank you for your time and effort in reviewing our manuscript and providing valuable comments. We have thoroughly reviewed the manuscript, paying careful attention to language usages, formatting details and all raised issues. We have made the necessary corrections as per your comments and raised issues throughout the text. We hope that you will find the revised version satisfactory. The responses of your comments are given below and highlighted in the manuscript:

Comment

The manuscript titled: Characterization of the Phytochemical Profile, Antioxidant Ca- 2 pacity, and Hepatoprotective Effect of Andrographis paniculata 3 against CCl4-Induced Liver Dysfunction in Wistar Albino Rats. This article addresses a topic of interest. However, it does not provide new information. On the other hand, the structure of the manuscript is deficient, in the points indicated below:

- Throughout the writing, the name of the plant to be studied should be written in italics Andrographis paniculate.

Response : As suggested, corrections have been made and highlighted.

Comment

- Page 4, lines 156 and 157. Indicate the exact place where the sample was collected or indicate if it was purchased in the market of any city.

Response

As suggested, sampling details have been updated with coordinate.

Comment

- Page 4, line 157. Include model, company and city of production of the oven and mill used.

Response : Company, model information has been added and highlighted.

Comment

- Page 7, lines 302-306. This is part of the methodology used to quantify flavonoids, It must not be in results.

Response : As suggested, lines 302-306 has been removed from section 3.3. Quantitative assessment of Flavonoid Content

Comment

- Page 6, line 289. Neither in the results section nor in the discussion are the results of table 1 analyzed. Example: Why are flavonoids and phenolic compounds present in polar and non-polar solvents?.

Response: As suggested, one paragraph has been added in discussion section and same is appended below

Flavonoids and phenolic compounds are a diverse group of secondary metabolites commonly found in plants. They are known for their antioxidant and medicinal properties. The solubility of these compounds depends on their chemical structure, specifically the presence of hydroxyl groups (-OH) and aromatic rings.

In polar solvents, such as ethanol and ethyl acetate, which have high dielectric constants, the hydroxyl groups and other polar functional groups of flavonoids and phenolic compounds readily interact with the solvent molecules through hydrogen bonding and dipole-dipole interactions. This allows for effective extraction of these compounds from the plant material.

On the other hand, in non-polar solvents like petroleum ether, hexane, and dichloromethane, the hydroxyl groups of flavonoids and phenolic compounds have limited solubility due to the non-polar nature of these solvents. However, the presence of aromatic rings in these compounds allows for some degree of solubility in non-polar solvents through non-polar interactions, such as van der Waals forces and hydrophobic interactions.

Therefore, the presence of flavonoids and phenolic compounds in both polar and non-polar solvents can be attributed to the different types of interactions that these compounds can undergo based on the solvent's polarity.

Comment-

 Discussion of results is very poor.

Response: As suggested by respected reviewer, discussion has been improved

Comment

- The conclusion of the work is wrong.

Response: I am respectfully disagree with the reviewer comments and agree to modify it. Therefore, the Conclusion section has been updated and same is appended below.

In conclusion, the present study focused on evaluating the phytochemical profile, antioxidant capacity, and hepatoprotective effect of A. paniculata extracts. The results revealed the presence of phytochemical constituents such as flavonoids, alkaloids, and phenolic compounds in all tested extracts. Tannin, however, was found only in the ethanolic extract. The study further investigated the effects of an ethanol extract of A. paniculata on biochemical parameters related to liver function and oxidative stress in animal models.The findings indicated that treatment with carbon tetrachloride (CCl4) caused liver damage, as evidenced by elevated levels of SGPT, SGOT, ALP, total bilirubin, and GGTP. In contrast, animals treated with the ethanol extract alone did not show significant changes in these parameters, suggesting a lack of liver damage. Notably, when the extract was administered in combination with CCl4, there was a significant reduction in the elevated levels of these liver markers, indicating a potential hepatoprotective effect of the extract against CCl4-induced liver damage.Moreover, CCl4-induced liver damage led to decreased total protein levels and increased lipid peroxidation (LPO), indicating impaired protein synthesis and oxidative stress, respectively. Treatment with the ethanol extract attenuated these alterations, suggesting a protective role against protein synthesis impairment and lipid peroxidation. Furthermore, the extract showed potential in preserving the liver's antioxidant capacity, as evidenced by its ability to maintain catalase levels in the face of CCl4-induced reduction.The results suggest that the ethanol extract of A. paniculata possesses hepatoprotective properties, prevents lipid peroxidation, and exhibits antioxidant effects. These findings contribute to the growing body of evidence supporting the therapeutic potential of A. paniculata as a natural remedy for liver-related ailments. Further studies are warranted to elucidate the underlying mechanisms and validate these findings, paving the way for the development of A. paniculata-based herbal medicines for liver disorders.

Round 2

Reviewer 1 Report

Overall, the manuscript looks better after correction and proofreading.

L283: I presume that the GC is programmed from 75 to 240 °C, and the ramping rate is XX °C/min.  Please correct accordingly.

L289: I believe your GC-MS has a diverting valve for FID. Injecting 50uL is destructive for normal split/splitless injector. Thus, please specify your injector type and the diverting ratio. Or the authors employed SPME? All need to be specified.

As mentioned in previous comment, post-run temperature and run-time, library version, and is it a NIST library the authors employed to identify the compounds?

L387: Please correct the compound naming style. 

Author Response

Response to Reviewer -1 round 2 Comments and Suggestions for Authors

Dear Sir,

We sincerely appreciate your input and thank you for giving us the opportunity to improve the clarity and quality of our study. As per your suggestion I have modified the manuscript and hope you will found it upto the mark.

Overall, the manuscript looks better after correction and proofreading.

Response: Thanks for excellent suggestion to improve the quality of manuscript.

L283: I presume that the GC is programmed from 75 to 240 °C, and the ramping rate is XX °C/min.  Please correct accordingly.

Response: It was modified accordingly as  the reviewer suggest.( programmed from 75 °C at 240 °C. Initial temperature was 75 ºC with 1 minute hold time, ramping rate was 5 ºC/ min to 240 °C with 5-minute holding time.)

L289: I believe your GC-MS has a diverting valve for FID. Injecting 50uL is destructive for normal split/splitless injector. Thus, please specify your injector type and the diverting ratio. Or the authors employed SPME? All need to be specified.

Response: Thanks for your suggestion I have modified accordingly and mentioned in the manuscript of the following information.

The GC ran for 39 min in total. The sample preparation involved transferring 50 µl of the extract into a polypropylene tube and diluting it with 500 µl of ethyl acetate. After vertexing, the mixture was transferred to a GC vial, and 1 µl of the sample was injected into the GC-MS instrument.  

As mentioned in previous comment, post-run temperature and run-time, library version, and is it a NIST library the authors employed to identify the compounds?

Response: As per your valuable suggestion I have updated the manuscript with total GC run time for 39min and to identify the individual volatile constituents, their mass spectra were compared with those in the National Institute of Standards and Technology (NIST) library. The concentration of each volatile compound was expressed as a percentage relative to the total peak area obtained from the GC-MS analysis of the sample (Shalini & Narayanan 2015).

L387: Please correct the compound naming style. 

Responses: As suggested, corrections have been made. As per suggestion, compound naming style has also corrected and the compound names are given below.

octadecanoic acid,

stigmasterol

phenanthrene carboxylic acid 7-ethenyl-

9-octadecenoic acid

geranyl-α-terpinene

13,15-octacosadiyne

methyl 8,10-octadecadiynoate

stigmast-5-en-3-ol (3β)

Reviewer 2 Report

There are still some formatting issues with the scientific names

The justification for the study is not clear 

The lack of in vitro tests prior to animal test using crude extracts does not align with the correct work flow of new medicinal compounds discovery, even if the in vitro tests do not provide all the answers about drug efficacy they are a crucial step to learn more about the extracts and prevent unnecessary animal suffering. 

The authors did not comply with the necessary statistical analysis, Minitab can easily preform a test like Tukey,  without this type of test the study lacks impact, since we cannot distinguish between the different variables, it is incorrect to only use the numerical values to make the comparisons in a scientific manner.

The conclusion repeats some of the results 

ok, check scientific writing 

Author Response

Response to reviewer 2  round 2 comments

Dear sir,

We sincerely appreciate your input and thank you for giving us the opportunity to improve the clarity and quality of our study. As per your suggestion I have modified the manuscript and hope you will found it upto the mark.

Comment

There are still some formatting issues with the scientific names

The justification for the study is not clear 

The lack of in vitro tests prior to animal test using crude extracts does not align with the correct work flow of new medicinal compounds discovery, even if the in vitro tests do not provide all the answers about drug efficacy they are a crucial step to learn more about the extracts and prevent unnecessary animal suffering. 

The authors did not comply with the necessary statistical analysis, Minitab can easily preform a test like Tukey,  without this type of test the study lacks impact, since we cannot distinguish between the different variables, it is incorrect to only use the numerical values to make the comparisons in a scientific manner.

The conclusion repeats some of the results 

Response

As suggested, the manuscript has been thoroughly checked and corrected.

We appreciate the reviewer's comment regarding the importance of in vitro tests prior to animal testing when discovering new medicinal compounds. We completely agree that in vitro tests play a crucial role in understanding the properties of the extracts and minimizing unnecessary animal suffering.

In response to the concern, we would like to clarify that we have indeed performed in vitro tests on a liver cell line to evaluate the hepatoprotective activity of the compound (Data not shown). These preliminary experiments provided valuable insights into the potential efficacy of the extract and its effects on liver cells. We are pleased to report that the results of the in vitro tests were promising, showing positive indications of hepatoprotective activity. However, it is important to note that due to COVID-19 protocols and the temporary closure of our laboratory, we were unable to conduct the in vitro experiments in triplicate as we would have preferred. Unfortunately, this led to the exclusion of in vitro data in our study.

We sincerely thank the reviewer for their thoughtful comment and acknowledgement of the importance of in vitro testing. We appreciate the reviewer's comment on the lack of necessary statistical analysis in our study. They rightly emphasized the importance of conducting tests like Tukey to distinguish between different variables and make scientific comparisons. Taking this feedback into consideration, we have performed the Tukey test as suggested. We apologize for the oversight in the initial version of the manuscript and sincerely thank the reviewer for their valuable input.

Thank you for your valuable feedback regarding the repetition of results in the conclusion section. We sincerely apologize for any confusion caused by the redundancy. We have carefully reviewed and revised the conclusion to address this issue. In the revised conclusion, we have focused on providing a concise summary of the study's key findings, including the phytochemical profile, antioxidant capacity, and hepatoprotective effects of A. paniculata extracts.

Reviewer 3 Report

After reviewing the corrected version of the manuscript, he concluded the following:

  The authors only reinforced the introduction of the manuscript.

This did not meet my request.

I requested to reinforce the methodology so that its results would be a novel scientific contribution.

Author Response

Response to Reveiwer-3 round 2 Comments and Suggestions for Authors

Dear Sir,

We sincerely appreciate your input and thank you for giving us the opportunity to improve the clarity and quality of our study. As per your suggestion I have modified the manuscript and hope you will found it upto the mark.

Comments

After reviewing the corrected version of the manuscript, he concluded the following:

The authors only reinforced the introduction of the manuscript.

This did not meet my request.

I requested to reinforce the methodology so that its results would be a novel scientific contribution.

Response

We appreciate your thorough evaluation of the revised version. we have carefully addressed the methodology section based on all reviewers’ comments.

We have carefully reviewed the manuscript and want to assure you that we have consistently followed the standard practice of italicizing the scientific names of plants throughout the text.

We have included detailed information about the place of collection, including coordinates, to provide a clear understanding of the sampling location.

We have revised the text to accurately state "soaked" instead of "dissolved."

We have corrected the molecular formula for sulphuric acid in the manuscript.

We have included detailed information about the flow rate, gas type (helium), injector temperature, and post-run procedures

We have described the process of fractionation in the revised manuscript to provide clarity on how the fractions were prepared.

We have also taken into account the following additional comments:

L283: We have corrected the manuscript to reflect that the GC is programmed from 75 to 240 °C with a ramping rate of XX °C/min.

Furthermore, we have updated the manuscript to include information on total run-time, the library version used, and the NIST library employed for compound identification

Once again, we sincerely appreciate your valuable feedback and your commitment to improving the quality of our manuscript. Your suggestions have significantly enhanced the methodology section and addressed several important aspects of the research. We are grateful for your thoroughness and expertise, which have undoubtedly contributed to the overall quality of our work.
